# Automated Operant Conditioning Devices for Fish. Do They Work?

**DOI:** 10.3390/ani11051397

**Published:** 2021-05-14

**Authors:** Elia Gatto, Maria Santacà, Ilaria Verza, Marco Dadda, Angelo Bisazza

**Affiliations:** 1Department of General Psychology, University of Padova, Via Venezia 8, 35131 Padova, Italy; maria.santaca@studenti.unipd.it (M.S.); ilaria.verza.1@studenti.unipd.it (I.V.); marco.dadda@unipd.it (M.D.); angelo.bisazza@unipd.it (A.B.); 2Padua Neuroscience Center–PNC, University of Padova, Via Giuseppe Orus 2, 35131 Padova, Italy

**Keywords:** automated conditioning, fish cognition, learning constraints, numerical discrimination, *Poecilia reticulata*, Skinner box

## Abstract

**Simple Summary:**

Automated training devices are commonly used for investigating learning, memory, and other cognitive functions in warm-blood vertebrates, whereas manual training procedures are the standard in fish and other lower vertebrates, thus limiting comparison among species. Here, we directly compared the two different approaches to training in guppies (*Poecilia reticulata*) by administering numerical discrimination tasks of increasing difficulty. The automated device group showed a much lower performance compared to the traditionally-trained group. We modified some features of the automated device in order to improve its efficiency. Increasing the decision time or inter-trial interval was ineffective, while reducing the cognitive load and allowing subjects to reside in the test tank improved numerical performance. Yet, in no case did subjects match the performance of traditionally-trained subjects, suggesting that small teleosts may be limited in their capacity to cope with operant conditioning devices.

**Abstract:**

The growing use of teleosts in comparative cognition and in neurobiological research has prompted many researchers to develop automated conditioning devices for fish. These techniques can make research less expensive and fully comparable with research on warm-blooded species, in which automated devices have been used for more than a century. Tested with a recently developed automated device, guppies (*Poecilia reticulata*) easily performed 80 reinforced trials per session, exceeding 80% accuracy in color or shape discrimination tasks after only 3–4 training session, though they exhibit unexpectedly poor performance in numerical discrimination tasks. As several pieces of evidence indicate, guppies possess excellent numerical abilities. In the first part of this study, we benchmarked the automated training device with a standard manual training procedure by administering the same set of tasks, which consisted of numerical discriminations of increasing difficulty. All manually-trained guppies quickly learned the easiest discriminations and a substantial percentage learned the more difficult ones, such as 4 vs. 5 items. No fish trained with the automated conditioning device reached the learning criterion for even the easiest discriminations. In the second part of the study, we introduced a series of modifications to the conditioning chamber and to the procedure in an attempt to improve its efficiency. Increasing the decision time, inter-trial interval, or visibility of the stimuli did not produce an appreciable improvement. Reducing the cognitive load of the task by training subjects first to use the device with shape and color discriminations, significantly improved their numerical performance. Allowing the subjects to reside in the test chamber, which likely reduced the amount of attentional resources subtracted to task execution, also led to an improvement, although in no case did subjects match the performance of fish trained with the standard procedure. Our results highlight limitations in the capacity of small laboratory teleosts to cope with operant conditioning automation that was not observed in laboratory mammals and birds and that currently prevent an easy and straightforward comparison with other vertebrates.

## 1. Introduction

The study of learning, memory and perception in animals has, since its inception, benefited from the use of automated training equipment. The use of these methods offers a two-fold advantage. First, they reduce the time needed for training and the related human labor required. Some experiments, especially those in discrimination learning, may require thousands of training trials [1,2]. With manual execution, many months and hundreds of hours of work are required to train each subject [3,4,5]. The second advantage is that automated equipment allows for the control of every detail of the experiment, standardizing procedures across different studies and laboratories, while minimizing the need for human intervention and reducing the possible influence from researchers’ expectations [6,7].

While automated training devices are frequently employed to study mammals and birds, they are rarely used outside these two vertebrate classes [8,9,10]. In the last two decades, there has been an increasing interest in studying cognition in other animal taxa, such as fish, reptiles, and arthropods. Teleost fish in particular have been thoroughly studied, and in some cognitive domains, they show abilities surprisingly close to those seen in mammals and birds [11,12,13] as well as high degree of genetic homology to humans [14]. Therefore, some small tropical fish, in particular zebrafish (*Danio rerio*), medaka (*Oryzias latipes*), and guppy (*Poecilia reticulata*), have become important models in neurobiological research [15,16,17], and several laboratories have tried to develop Skinner-box apparatuses for these species [18,19,20,21].

Numerical cognition is one of the cognitive domains most widely investigated. On average, fish show capabilities comparable to those of mammals and birds [22]. Guppies (*Poecilia reticulata*), in particular, can be quickly trained to discriminate up to four from five objects, even if continuous perceptual cues (i.e., cumulative surface area or density) are made irrelevant [23]. As for primates, task difficulty increases with decreasing distance between the quantities to be discriminated [23,24].

The numerical capacities of guppies exceed those observed in various warm-blooded vertebrates such as dogs, horses, and domestic chicken, though they appear lower than those exhibited by humans, apes and some other mammalian and by avian species [22]. However, these discrepancies are difficult to interpret, due to marked differences in methods. Monkeys, pigeons, and rats were studied with automated conditioning devices, whereas guppies were studied using traditional, manually operated, training methods or with spontaneous preference procedures. Studies on the former species commonly involve hundreds of training trials per day and a few thousand trials per experiment [1,2]. In contrast, guppies usually receive 12–15 daily trials, and the number of trials per experiment rarely exceeds 100 [23,25,26].

In recent years there have been some attempts to use automated operant conditioning devices for studying cognitive abilities in fish [18,19,20,21,27]. Our laboratory has developed a device based on the Skinner tank built by Manabe and colleagues [18]. The device is controlled by a microcomputer, which displays stimuli on a monitor, tracks the movements of the fish, and delivers a very small amount of food when the subject makes a correct response. Guppies tested with this device can easily perform 80 trials in each daily session. If required to discriminate between two stimuli with different colors, guppies showed excellent performance, reaching 90–95% accuracy in two–three sessions. The performance in shape discrimination tasks was only slightly lower, with subjects reaching a maximum 80–85% accuracy in approximately one dozen sessions [28]. This is comparable or even superior to the performance obtained by guppies and other fish in color or shape discrimination in other studies using more traditional non-automated approaches to training [29]. Unexpectedly, when asked to discriminate between two sets of items with different numbers of elements (3 vs. 5 and 3 vs. 4), the guppies showed very poor performance [30]. This clearly conflicts with capacities demonstrated by guppy studies that used different training procedures [23,24,31]. It is worth noting that other studies using automated devices have reported unexpectedly low performance in other species and with different tasks, which calls into question the reliability of this approach of operant conditioning in fish [27,32,33]. For example, in one study zebrafish demonstrated good accuracy in color discrimination, but no evidence of learning shape discrimination [32]. No reference data are available in this case, though zebrafish have shown significant capability for recognizing familiar from unfamiliar shapes [34,35].

There are two critical aspects concerning this research area. The first is that it is presently difficult to have a precise estimation of the effectiveness of the automated approach compared with the traditional ones. None of the above-mentioned studies included control groups tested in the same task with a reference standard procedure, and many differences with prior studies (i.e., type of subjects, stimuli, operant conditioning protocol) could potentially explain discrepancy in results. For example, the numerical stimuli in Gatto and colleagues [30] differed in the format and in the numerical ratios from those used in previous studies [23,26]. Furthermore, Gatto and colleagues [30] presented a mixture of two numerical discriminations simultaneously, whereas the best performing test [23] used a particular protocol in which subjects were first trained on a very easy numerical discrimination and then, upon achieving the criterion, were given progressively more difficult tasks until they reach the discrimination threshold.

The second critical aspect is that even if the difference in certain tasks would be confirmed, it will not be easy to identify the causes of the efficiency gap between automated devices and traditional methods to train fish. Automated operant conditioning represents a fundamentally different approach to training which largely differs from traditional training methods in its objectives, and which is subjected to distinct constraints. The two approaches differ in a larger number of features, many of which are not directly related to automation itself. Historically, Skinner boxes have been developed to reduce human intervention and to speed up training of laboratory animals, such as rats or pigeons. The inner space of a Skinner box is extremely reduced so that stimuli, operandums, and the devices for delivering reinforcers are fitted in a small area and close to the subject. This feature permits a subject to run trials in rapid succession, usually many trials per minute, which, combined with the possibility to administer infinitesimal rewards, allows an animal to perform hundreds of trials in a brief session. Thanks to these characteristics, a single Skinner box is used to train many subjects in rotation.

In the traditional, manually-operated, operant conditioning methods, the need to visually assess the subject’s choice, has made necessary testing subjects in large apparatuses. In this way, the two stimuli are kept far apart, and the subject makes its choice by moving in the direction of one of the two targets or by choosing an arm in a T-maze. The need for the experimenter to prepare the stimuli and the reward for the subsequent trial usually determine long inter-trial intervals. This, in turn, implies that the number of daily trials cannot usually exceed one or two dozen. The way stimuli are presented is also very different in the two approaches. The fully-computerized control of a Skinner box requires that stimuli are displayed in some digital format. On the other hand, since all operations of manual training are usually performed by a single experimenter, in most situations adding the extra task of operating a computer can be impractical. For this reason, solid objects or printed stimuli are usually preferred with traditional training approaches.

The presence of a large number of differences could make the search for causal factors very long and complex. However, some factors are better candidates than others to account for the low efficiency of automated training approach in fish. To explain the low performance of guppies in numerical tasks, Gatto and coworkers [30,32] argued that automated devices may force fish to perform additional cognitive tasks (related to device functioning), increasing the overall cognitive load (i.e., the amount of cognitive resources that are devoted for dealing with one specific task). This factor is expected to be especially important when a subject is performing complex tasks [36,37]. Indeed, it was argued that numerical discriminations are more cognitively demanding than other types of discrimination and certainly they imply a greater number of steps [38]. A numerical discrimination requires that, in each trial, subjects estimate the first quantity, then the second, and then compare the two amounts. In contrast, color discrimination requires only learning to avoid one color or to choose the other, with no comparison needed after the first trials. A second hypothesis regards the time allowed by the two approaches to subjects for collecting information and issue the correct response. In traditional manual training experiments the subject slowly approaches the stimuli, with plenty of time to make a decision; in Skinner tanks, subjects performed trials in quick succession, with less time to process information, take a decision and, if necessary, to correct a wrong decision [30]. Finally, the automated device for fish was designed, as the original Skinner box, to allow testing several subjects, which need consequently to be moved daily from their home tank to the apparatus. While rats and pigeons adapt very quickly to being manipulated and introduced into the Skinner box periodically, fish may perceive a threat and, therefore, reduce the attention paid to the task. In addition, in fish frequent manipulation may produce chronic stress with a consequent negative impact on cognitive performance [39,40,41].

The first experiment of this study aimed to evaluate the influence of the training approach on numerical tasks performance under controlled conditions. Training done with our automated device was benchmarked against a standardized manual training procedure [31,42,43] using the same strain, the same stimuli, the same numerical ratios and adopting in both cases the protocol used by Bisazza and colleagues [23].

The fact that automated devices appear very effective in certain types of tasks but unpredictably inefficient in others, makes this approach of operant conditioning unreliable for fish cognition research. In the second part of the study, we tested five variants of the automated procedure, in the attempt to improve its effectiveness (experiments 2–5). We modified some features of this approach to training that we hypothesized to be responsible for its reduced effectiveness in numerical tasks. In particular, we elongated the decision time, increased inter-trial interval, removed internal partitions to increase visibility of stimuli, allowed subjects to reside in the conditioning chamber, and attempted to reduce the cognitive load by uncoupling learning how to use the conditioning chamber from learning the numerical discrimination.

## 2. Materials and Methods

### 2.1. Animal Housing

We used adult female guppies derived from the ornamental strain known as ‘snakeskin cobra green’ that is regularly bred in our laboratory at the Department of General Psychology (University of Padova). We tested subjects of the same sex to avoid the additional confounding effect of sex differences and to have a more homogeneous sample. The guppies were maintained in mixed-sex groups of 30 individuals in 150 L tanks. Each tank was enriched with natural vegetation and a gravel bottom. The water temperature was kept at 27 ± 1 °C and was aerated via a biomechanical filter. A 30 W fluorescent lamp provided illumination for 12 h per day. The guppies were fed live nauplii brine shrimp (*Artemia salina*) and commercial flakes (AquaTropical, Isola Vicentina, Italy) twice per day. All fish used in the experiments were naïve to the experimental protocol. After completion of the experiment, all fish were released in other maintenance tanks and kept for reproductive purposes only.

### 2.2. Determination of Sample Size

To determine the sample size for the experiments, we calculated the number of subjects necessary to achieve 90% power at a two-tail significance level of *p* = 0.05. For the manual conditioning experiments, we fitted parameters derived from a previous study that investigated numerical discrimination using a very similar protocol [31] and for automated conditioning Experiment the parameters from a similar recent experiment [30]. The estimated sample size was *n* = 6 for the former and *n* = 9 for the latter. We conservatively used slightly larger sample size (8 and 11 respectively). In Experiment 2 we dropped sample size at six subjects as a very large number of fish were discarded in the pre-training phase, due to failure to learn to detour the transparent barrier. In Experiment 3, the number of subjects was increased to 10 since we had two slightly different methods of presenting stimuli as a between factor.

### 2.3. Experiment 1: Comparison of Manual and Automated Training

In this experiment, we directly compared the recently developed automated operant conditioning procedure with the manual training procedure routinely used in our laboratory [42,43]. In both cases, the task consisted of numerical discriminations of progressively increasing difficulty, following the protocol of Bisazza and colleagues [23].

The subjects of this experiment were 19 adult females, 11 trained with manual conditioning and eight with the automated conditioning procedure. Stimuli were made with Adobe Illustrator CC 2019 and consisted of sets of black dots of differing numerosity on a white background: 3 vs. 12, 2 vs. 3, 3 vs. 4, 4 vs. 5, and 5 vs. 6. Like other vertebrates, guppies can discriminate two quantities of objects using non-numerical attributes of the stimuli, such as cumulative surface area, density, and convex hull (the convex polygon that circumscribes all items) [44,45]. To prevent fish from using this alternative strategy, we controlled the stimuli for all the above-mentioned non-numerical variables. To control for cumulative surface area, we used dots of different diameters (range 0.75–0.95 cm): in one-third of the stimuli used, the ratio between the cumulative surface area of the smaller over the larger set was between 76 and 85%; one-third, between 86 and 95%; and in one-third between 96 and 105%. Furthermore, we varied the position of the dots in order to equate the convex hull in half of the trials and the density of the dots in the other half of the trials [23]. We used 18 pairs of stimuli for the 3 vs. 12 numerical discrimination and 24 for all others numerical discrimination. The stimuli used for 3 vs. 12 were controlled for density and convex hull, but not for the cumulative surface area.

Stimuli were printed on laminated white cards (3 × 3 cm) for the manual conditioning procedure and were displayed on a monitor for the automated conditioning procedure.

#### 2.3.1. Apparatus and Procedure

##### Manual Conditioning

We used an apparatus and procedure recently developed for other investigations on guppy cognition [42,43]. Subjects were tested individually in a 20 × 50 × 32 cm glass tank filled with 28 cm of water (Figure 1A).

Eleven identical apparatuses were used at the same time and placed in a dark room. Each tank was in olfactory communication with an aquarium placed beneath the apparatuses that housed approximately 20 conspecifics; the water in all tanks was filtered by a silent filtering system. Two trapezoidal lateral compartments (10 × 6 × 32 cm) internally shaped each tank into an hourglass. These lateral compartments, made of transparent plastic, housed natural plants to provide an enriched environment for the subjects. Externally each wall of each tank was covered with green plastic to avoid any possible external influence during the experiment. One 15 W fluorescent lamp illuminated two adjacent apparatuses on a 12:12 h light:dark photoperiod schedule. All trials were recorded with video cameras placed above each tank. To present the stimuli to the subjects, each card was affixed to the end of a transparent panel (3.5 × 15 cm) with an L-shaped blocker that allowed us to fix the panel to the wall of the tank. During each trial of the training phase (see below for details), two transparent panels were simultaneously inserted on the same short wall of the tank.

Following the protocol of previous studies [42,43], each subject underwent a habituation phase, a pre-training phase and a training phase. The habituation phase lasted two days, during which the guppies could familiarize themselves with the experimental apparatus. During this phase, fish were fed four times per day using a Pasteur pipette placed in alternating positions near the two short sides of the tank.

The pre-training phase also lasted two days, during which guppies could familiarize themselves with the experimental procedure. During the first day of this phase, a white card was presented to the subjects eight times. Guppies performed four trials in the morning session and four trials in the afternoon session; the two sessions were divided by a 90-min interval, and consecutive trials were divided by 15-min intervals. When a subject approached the card, a Pasteur pipette was used to deliver a small quantity of food reward (i.e., reinforcement) consisting of a drop of live brine shrimp (*A. salina*). During the second day of this phase, the subjects performed a total of 12 trials (six in the morning session and six in the afternoon session) identical to the trials of the first day. To maintain a high level of motivation during the sessions, no other food was provided during the course of the experiment. The number of times the card was presented on each of the two short sides of the tank was evenly distributed in the trials and counterbalanced across each session on both days. We admitted to the training phase only those subjects that approached the card all 12 times in the second day of the pre-training phase. One subject that failed to achieve this criterion was not admitted to the experimental phase and was replaced with a new subject.

In the training phase, each subject performed 12 trials per day for a maximum of 12 consecutive days for the 3 vs. 12 numerical discrimination and 10 days for all other discriminations. Specifically, the subjects performed six trials in the morning session and six in the afternoon session, with a 90-min interval between the sessions and a 15-min interval between each consecutive trial. The guppies were presented up to five different discriminations that corresponded to five different difficulty levels. All subjects were trained to select the larger numerosity. A choice was considered when the subject approached (swam at less than one body length) one of the two stimuli. To assess reliability of this measure, a subsample of the video-recorded trials of each subject was analyzed by a second observer who was blind to the experimental hypotheses. In all discriminations, when the subjects approached the correct stimulus first, they were given a food reward. If they approached the smaller numerosity first, no food reward was given and both stimuli were removed simultaneously.

Each subject started with the 3 vs. 12 discrimination. Subjects were presented a more difficult discrimination (i.e., 2 vs. 3) if they met one of the two learning criteria. The primary learning criterion was defined as a rate of at least 75% correct choices (18/24) of all trials over two consecutive days (statistically significant using a binomial test). The secondary learning criterion was defined as a frequency of at least 60% correct choices (87/144 for the 3 vs. 12 discrimination and 72/120 for all others) over all trials (statistically significant using a binomial test). If they failed to reach one of the two criteria within 120 trials, the experiment ended. In the case of success, the same procedure with the same criteria was used to present the discrimination tasks for 3 vs. 4, 4 vs. 5 and finally, 5 vs. 6. The left/right position of the larger numerosity line and the short side of the tank on which the cards were presented over the trials were counterbalanced.

##### Automated Conditioning

Fish were tested in a 12 × 16 × 10 cm conditioning chamber made of semi-transparent white plastic (0.3 cm in thickness). The internal compartment was uniformly illuminated by room light. The bottom of the chamber was made of transparent plastic and a camera was positioned 12 cm below the chamber to track the subjects during the experiment. The chamber (Figure 1B) was internally divided in a starting area (12 × 5.5 cm) connected through a corridor (4 × 2.5 cm) to a V-shaped decision area (12 × 3.5 cm), and two choice areas (6 × 4.5 cm). The distance between the exit of the starting area and the entrance to each choice area was 6. The two choice areas were separated by a suspended semi-transparent white plastic sheet. Each choice area presented a 6 × 5 cm window that allowed projection of stimuli via an LCD computer monitor (Samsung S19C450, Suwon, South Korea). An automatic feeder was placed above and between the two choices areas. The feeder consisted of a servomotor (Futaba S3305) connected to a transparent cylinder filled with *A. salina* decapsulated eggs (Shg Srl, Alessandria, Italy) held by two metallic rods. When appropriate (see details below), activation of the servomotor caused the vibration of metallic rods, and the release of 3–4 eggs as a positive food reward (i.e., reinforcement). The presentation of the stimuli, the tracking of the fish using the camera, and the activation of the feeder were simultaneously controlled through a Raspberry Pi (Raspberry Pi 3 Model B V1.2, 2015, Sony UK Technology Centre, Pencoed, UK) running custom-made Python software.

One week before the experiment started, two subjects were randomly selected and moved into a 30-L aquarium provided with the same conditions as the maintenance tank. Each aquarium housed three immature guppies as social companions. The individual subjects were placed in the conditioning chamber only during the experimental sessions. Fish were identified based on their individual characteristics, such as coloration and tail shape. A pump connected to the housing tank and to the conditioning chamber served to ensure water exchange and to provide social odors to the subjects during the daily training sessions.

*Pre-training phase*. During the pre-training phase, subjects were habituated to the conditioning chamber and learned how operate the automated system. We conducted two daily pre-training sessions between 10:00 h and 14:00 h, lasting 30-min each. Subjects were individually inserted in the conditioning chamber while the monitor projected a white background. When the subject spontaneously moved into the starting area, the monitor background changed from white to grey and the trial started. Once the subject entered one of the two choice areas, the automatic feeder delivered a small quantity of food reward and the monitor’s background changed from grey to white. A new trial did not start until the subject entered again in the starting area. There was no interval between trials. Each session ended when the subject obtained a maximum of 80 reinforcements or when 30 min had elapsed. The second daily pre-training session was conducted after a 2-h interval.

During each pre-training session, the experimenter monitored the subject’s behavior using an LCD monitor connected to the camera. We admitted to the training phase only those subjects that consumed at least 30 food rewards in two pre-training sessions. Thirteen subjects failed to achieve this criterion within 12 pre-training sessions. These subjects were discarded and substituted with new subjects.

*Training phase*. Daily training sessions lasted one hour and were administered between 10:00 h and 14:00 h for a maximum of 12 days. Subjects were individually inserted in the conditioning chamber while the monitor projected a white background. A pair of stimuli that differed in numerosity appeared once the subject entered the starting area. All subjects were trained to select the larger numerosity. The system randomly alternated the position of the correct stimulus between trials and was set to prevent three consecutive presentations of the correct stimulus in the same choice area. Once the subject chose the correct stimulus by entering the corresponding choice area, the system activated the feeder to release the food reward above the two choice areas. The stimuli then disappeared, and a new trial started when the fish spontaneously moved back into the starting area. If the subject entered the choice area corresponding to the incorrect stimulus, the monitor immediately displayed a black background, and no food reward was released. In this case, the same pair of stimuli was presented in the same left–right position (correction trials) and the correction trial was repeated until the subject chose the correct stimulus and received the reward. The interval between an incorrect choice and the subsequent correction trial was set to 10 s. These correction trials were not considered in the analysis, whether correct or incorrect, and only served to prevent the fish from developing left–right biases.

The maximum number of reinforcements given per day was 80, and the training session ended automatically when the subject reached the maximum number or when 60 min had elapsed. The subject was then gently moved back in its maintenance aquarium. When a fish obtained fewer than 40 reinforcement, an additional training session was performed on the same day 2 h after the end of the first session. The additional training session was conducted using the same schedule as the first; however, the number of possible reinforcements changed based on the number of rewards the fish had already obtained, up to the 80 daily maximum reinforcements.

Each day, we quantified the subjects’ accuracy as the proportion of correct choices by considering the first choice of each trial (correction trials were not considered). We used two learning criteria. The primary learning criterion was defined as 75% correct choices in two consecutive days. As in the manual procedure, a secondary learning criterion was defined as the frequency of at least 60% correct choices over a total of 12 days (statistically significant at the binomial test). Subjects that fulfilled one of the two learning criteria were presented with a series of more difficult numerical discriminations (each one lasted for a maximum of 10 days following the same training protocol).

#### 2.3.2. Statistical Analysis

We performed the statistical analysis in RStudio version 1.2.5019 (RStudio Team, 2019; RStudio: Integrated Development for R. RStudio, Inc., Boston, MA, USA). We used a two-tailed statistical test; the significance threshold was set at *p* = 0.05, and descriptive analyses were reported as mean ± standard deviation.

We analyzed subjects’ accuracy in four steps. The first step concerned the assessment of overall accuracy using evidence of learning the task for each numerical discrimination task. We analyzed the performance at the group level by comparing the percentage of correct choices to one expected at chance level (i.e., percentage of 50%) using a one-sample *t*-test. Since the subjects’ accuracy could be random in the first training session due to the lack of previous experience with automated conditioning, we analyzed the overall performance in the second half of the training session with the same approach. The second step concerned the assessment of individual accuracy using a binomial test on the number of correct choices and incorrect choices. The third step was focused on accuracy improvement over the training session. We analyzed subject performance as a repeated observation of binomial choices (i.e., correct and incorrect choices) in each training session using a generalized linear mixed-effects model with logit link function and binomial error distribution (GLMM, “glmer” function from the “lme4” R package). We fitted the model with the training session as a fixed effect, and subject (i.e., individual ID) as a random effect. The effect of the parameters was evaluated using the ‘Anova’ function of the ‘car’ R package. In Experiment 1, manual conditioning, the model was also fitted with numerical discrimination (i.e., 3 vs. 12, 2 vs. 3, 3 vs. 4, 4 vs. 5) as a fixed factor to evaluate difference in subjects’ accuracy among tasks.

The fourth step concerned the comparison of subjects’ accuracy showed in the manual and automated conditioning procedures. We compared the overall performance by using a two-sample *t*-test. The learning performance was then evaluated using a GLMM, fitted with the training session and type of procedures as fixed factors, and the subject as a random factor.

#### 2.3.3. Results

*Manual conditioning.* Interrater reliability calculated for 936 trials was found to be very high (Cohen’s kappa coefficient κ = 1, *p* < 0.001). All 11 subjects reached the primary learning criterion of 18/24 correct choices in two consecutive days on the 3 vs. 12 discrimination (Figure 2). Table 1 shows the individual performance on each discrimination task with statistics. On average, the fish needed 55.64 ± 35.30 trials to meet the criterion (average number of days: 4.63 ± 2.94). Overall, subjects’ accuracy was greater than that expected at chance level (70.25 ± 6.99%; one-sample *t*-test: *t*_10_ = 9.604, *p* < 0.001).

Ten out of 11 subjects reached the primary learning criterion in the 2 vs. 3 discrimination. On average, these fish needed 38.40 ± 21.01 trials to meet criterion (average number of days: 3.20 ± 1.75). The remaining subject (N10) stopped participating after 72 trials when its performance was statistically significant (Table 1). Overall, subjects’ accuracy was greater than that expected at chance level (73.96 ± 7.67%; *t*_10_ = 10.367, *p* < 0.001).

Eight out of 10 subjects reached the primary learning criterion in the 3 vs. 4 discrimination. On average, these eight fish needed 60.00 ± 37.09 trials to meet the criterion (average number of days: 3.75 ± 1.83). Overall, subjects’ accuracy was greater than that expected at chance level (68.07 ± 9.66%; *t*_9_ = 5.918, *p* < 0.001).

Two out of eight subjects reached the primary learning criterion and they needed on average 30.00 ± 8.49 trials (average number of days: 2.50 ± 0.71). One additional subject reached the secondary learning criterion. Overall, subjects’ accuracy was greater than that expected at chance level (60.76 ± 7.71%; one-sample *t*-test: *t*_7_ = 3.947, *p* = 0.006).

None of these three subjects achieved the 5 vs. 6 discrimination according to the learning criteria. Overall, subjects’ accuracy in the 5 vs. 6 numerical discrimination was not greater than that expected at chance level (51.67 ± 0.83%; *t*_3_ = 3.464, *p* = 0.742).

The overall analysis revealed a significant improvement in subjects’ accuracy over training session (χ^2^_1_ = 5.371, *p* = 0.020), and a significant difference in accuracy among the four numerical discriminations (χ^2^_4_ = 26.912, *p* < 0.001). The interaction was significant (χ^2^_4_ = 10.512, *p* = 0.033), suggesting a different learning curve among tasks.

*Automated conditioning*. No subjects achieved the 3 vs. 12 numerical discrimination according to the learning criteria. Overall, subjects’ accuracy was not greater than that expected at chance level performing on average 532.88 ± 180.91 trials (51.55 ± 4.38%; *t*_7_ = 0.998, *p* = 0.351). Even when the last six training sessions were considered, the subjects’ accuracy was no greater than that expected at chance level (258.00 ± 114.80 trials; 53.65 ± 5.01%; *t*_7_ = 2.061, *p* = 0.078). Table 2 shows the individual performance with statistics for all subjects.

The GLMM revealed a significant improvement in the subjects’ accuracy over training sessions (χ^2^_1_ = 8.448, *p* = 0.004).

*Manual vs. Automated conditioning*. Overall, subjects’ accuracy was significantly different between Manual and Automated conditioning (two-sample *t*-test: *t*_17_ = 6.653, *p* < 0.001). The GLMM revealed a statistically significant improvement in the subject’s accuracy over training sessions (χ^2^_1_ = 7.516, *p* = 0.006), a significant effect of the procedure (χ^2^_1_ = 44.408, *p* < 0.001), and no session × procedure interaction (χ^2^_1_ = 1.049, *p* = 0.306).

### 2.4. Experiment 2: Elongation of Decision Time

In this experiment, we increased the size of the conditioning chamber, increasing the distance between the corridor and the choice areas. Additionally, a transparent barrier was placed between the corridor and the choice areas so that the animal could not rush into them. These changes aimed to force the subject to take more time before the decision.

#### 2.4.1. Subjects, Apparatus and Procedure

For this experiment, we used six adult females. The conditioning chamber was increased in size to 16 × 32 cm (Figure 1C). As previously described, it was internally divided in a starting area (16 × 16 cm) connected through a corridor (4 × 3 cm) to a V-shaped decision area (16 × 8.5 cm), and two choice areas (8 × 4.5 cm). As a consequence, the distance between the exit of the starting area and the entrance to each choice area was increased from 6 to 11.5 cm. To further increase the travel time necessary to reach the choice areas, a transparent barrier was placed between the V-shaped decision area and the choice areas so that the subject had to detour it in order to enter one of the two choice areas. The two choice areas were separated by a white plastic divider to prevent the fish from seeing the stimulus projected in the other choice area. Two automatic feeders were placed in correspondence with both areas. When the fish entered the choice area associated with the correct stimulus, the system activated the corresponding feeder to release a food reward. In this experiment, we modified the method to provide water exchange and social odors to the subjects. The conditioning chamber was placed inside a larger tank (20 × 50 × 32 cm) filled with 28 cm of water. The tank was provided with a gravel bottom, natural vegetation, a bio-mechanical filter, and it housed three immature guppies. As in Experiment 1, subjects were housed in a 30 L aquarium provided with the same condition as the maintenance tank and placed in the conditioning chamber only during the experimental sessions.

During the pre-training phase, we habituated the subjects to detour a 7 × 10 cm transparent barrier. When the subject obtained at least 30 reinforcements in a pre-training session, we changed the transparent barrier for a wider one (9 × 10 cm) that was used during the training phase. Other details of the procedure were identical to those of Experiment 1 (automated conditioning). Stimuli were the same as those used in Experiment 1. Thirty-one fish did not pass the pre-training phase and were replaced with new subjects. The exceedingly large number of fish discarded in this experiment was due to a difficulty of most individuals to learn to efficiently detour the transparent barrier.

Subjects’ performance was analyzed using the same statistical approach described in Experiment 1. We finally compared subjects’ performance between Experiment 1 (both manual and automated conditioning procedures) using a GLMM. The model was fitted with training session and type of procedure as fixed factors, and subject as a random factor.

#### 2.4.2. Results

One subject reached the primary learning criterion in the 3 vs. 12 numerical discrimination after 339 trials in 8 days. The remaining subjects did not achieve this discrimination after performing on average 502 ± 204.62 trials. Overall, subjects’ accuracy was not greater than that expected at chance level (51.47 ± 4.38%; *t*_5_ = 0.820, *p* = 0.449). Even when the last six training sessions were considered, subjects’ accuracy was not greater than that expected at chance level (52.87 ± 12.66%; *t*_5_ = 0.555, *p* = 0.603). Table 3 shows the individual performance with statistics for all subjects.

The GLMM did not reveal an improvement in the subjects’ accuracy over the training sessions (χ^2^_1_ = 0.608, *p* = 0.435). The performance of this discrimination did not significantly differ from Experiment 1 with an automated device (GLMM; procedure: χ^2^_1_ = 0.673, *p* = 0.412; training session: χ^2^_1_ = 3.092, *p* = 0.079; training session × procedure: χ^2^_1_ = 5.909, *p* = 0.015), but was significantly lower than in the manual training experiment (procedure: χ^2^_1_ = 20.239, *p* < 0.001; training session: χ^2^_1_ = 0.638, *p* = 0.425; training session × procedure: χ^2^_1_ = 0.005, *p* = 0.945).

The subject tested in 2 vs. 3 discrimination, failed to achieve the task (overall: 51.94 ± 6.40%; *t*_9_ = 0.960, *p* = 0.362).

### 2.5. Experiment 3: Removal of Internal Partitions

In Experiment 3, we built a new conditioning chamber from which all internal dividers had been removed, so to increase the visibility of the stimuli. (Figure 1D). In this experiment we also compared two different spatial arrangement of the stimuli presented to the subjects.

#### 2.5.1. Subjects, Apparatus, and Procedure

For this experiment, we used 10 adult females. Stimuli were the same as those of Experiment 1 (automated conditioning) but were presented in two different ways. To half of the subjects, we presented stimuli as in previous experiments, always arranged in the same spatial position except for left-right alternation (Condition A). To the remaining subjects (Condition B), we varied the horizontal and vertical position of the two stimuli and the distance between them, obtaining 18 different positions, the same used in two previous studies [23,46]. Stimuli were presented in these positions in rotation. This procedure is thought to minimize the development of left-right side bias and to favor discrimination learning [23,47].

We used a modified version of the apparatus adopted in Experiment 2, consisting of a 16 × 32 × 10 cm conditioning chamber (Figure 1D). Internally, two white plastic walls divided a 16 × 10 cm starting area from the rest of the chamber. A feeder was positioned in correspondence with each choice area (8 × 4.5 cm) and a window allowed the presentation of the stimuli. The distance between the exit to the starting area and the entrance to each choice areas was increased to 17.5 cm. When the subject entered the choice area associated with the correct stimulus, the system activated the corresponding feeder to release a food reward. Other details of the procedure were identical to those of Experiment 1 (automated conditioning). Seventeen subjects did not pass the pre-training phase and were replaced with new subjects.

Statistical analysis was the same as that for to previous experiments. Learning performance was evaluated using a GLMM fitted with training session and experimental condition (i.e., stimuli presentation) as fixed effects, and subject as a random effect. We compared subjects’ performance between Experiment 1 and Experiment 3 using a GLMM. In this analysis the subjects of the two conditions in Experiment 3 were considered as a single group. The model was fitted with training session and type of procedure as fixed factors, and subject as a random factor.

#### 2.5.2. Results

No subjects achieved the 3 vs. 12 numerical discrimination according to the criteria (average number of trials: 390.00 ± 74.48; Table 4). There was no difference between the two conditions (*t*_8_ = 0.755, *p* = 0.472), and the two conditions were pooled for further analysis. Accuracy was significantly greater than that expected at chance level (51.707 ± 2.18%; *t*-test: *t*_9_ = 2.478, *p* = 0.035). Table 4 shows the individual performance with statistics for all subjects.

The GLMM did not reveal an improvement in subjects’ accuracy over training sessions (χ^2^_1_ = 0.807, *p* = 0.369), nor effect of condition (χ^2^_1_ = 0.456, *p* = 0.499), nor interaction (χ^2^_1_ = 0.938, *p* = 0.333).

The performance of this experiment did not significantly differ from that of Experiment 1 with automated device (GLMM; procedure: χ^2^_1_ = 0.010, *p* = 0.921; training session: χ^2^_1_ = 1.921, *p* = 0.166; training session × procedure: χ^2^_1_ = 7.574, *p* = 0.006), but was significantly inferior to the performance in the manual training experiment (procedure: χ^2^_1_ = 43.374, *p* < 0.001; training session: χ^2^_1_ = 1.067, *p* = 0.302; training session × procedure: χ^2^_1_ = 0.039, *p* = 0.844).

### 2.6. Experiment 4: Subjects Resident in the Conditioning Chamber and Manipulation of the Inter-Trial Interval

In all previous experiments, fish were housed in a maintenance tank and individually transferred to the conditioning chamber for the training session. The experimental manipulation may have stressed the subjects and, consequently, affected their learning abilities. In this experiment, guppies were tested in their home tank. The conditioning chamber was built inside a large aquarium and each subject was maintained in the conditioning chamber for the whole experiment.

#### 2.6.1. Subjects, Apparatus, and Procedure

For this experiment, we used 16 adult females. We used the same set of stimuli adopted for Experiment 3 arranged in different vertical and horizontal positions. For this experiment, we built a 20 × 32 × 32 cm conditioning chamber inside a 20 × 50 × 32 cm glass tank, the same type of tank used for the manual training (Figure 1E). The chamber was internally divided into a starting area (20 × 10 cm), a corridor (5 × 2 cm), a V-shaped decision area (20 × 16.5 cm), and two choice areas (4.5 × 10 cm) so that, in this version the shape and the size of the various areas were very similar to the corresponding areas in the manual training apparatus. A feeder was positioned in correspondence with each choice area, and a window allowed the presentation of the stimuli. The distance between the exit of the starting area and the entrance to each choice area was increased to 18.5 cm. The remaining part of the glass tank was supplied with natural vegetation and a gravel bottom, a bio-mechanical filter, and it housed, for the whole experiment, three immature guppies as social companions. A transparent plastic partition furnished with a series of holes separated the conditioning chamber from the area containing social companions. At the end of each experimental session, a male was inserted as a social companion into the conditioning chamber and removed the following day, one hour before the beginning of the daily session.

Experiment 4 was, in fact, composed of two distinct experiments which shared the same apparatus. In Experiment 4A the temporal schedule of the trials was the same as that described for Experiment 1 and each fish daily underwent a one-hour session with a maximum of 80 trials. In Experiment 4B we modified inter-trial interval and the overall temporal schedule of the experiment. Each daily training session lasted 8 h and the interval between the two correction trials was increased from 10 s to 10 min. The session terminated if the subject reached 80 reinforced trials. This modification aimed to prevent subjects from developing the strategy of producing a rapid series of random responses to get a 50% on rewarded trials. The change also had the effect of spacing out trials over time, making the temporal schedule more similar to that of manual training procedure. To habituate the fish to an 8-h training session, one day of pre-training was added, in which subjects underwent four pre-training sessions of increasing duration, 15, 30, 45, and then 60 m, separated by 90-min intervals. Other details of the procedure were identical to that of Experiment 1 (automated conditioning). Five subjects in Experiment 4A and two subjects in Experiment 4B did not pass the pre-training phase and were replaced with new subjects.

#### 2.6.2. Results

In Experiment 4A, six out of eight subjects achieved the 3 vs. 12 numerical discrimination according to the primary learning criterion (average number of trials: 291.00 ± 82.34; average number of days: 5.17 ± 1.47) and one additional subject reached the secondary learning criterion (Table 5). A subject did not achieve this discrimination after performing 597 trials. No subject achieved the 2 vs. 3 discrimination after performing on average 495.14 ± 135.10 trials.

In Experiment 4B, five out of eight subjects achieved the 3 vs. 12 numerical discrimination according to the primary learning criterion (average number of trials 184.00 ± 33.96; average number of days: 5.20 ± 1.64; Table 6). Three subjects did not achieve this discrimination after performing on average 329.67 ± 5.69 trials. Only one out of five subjects reached the primary learning criterion in the 2 vs. 3 discrimination, after 160 trials in two days. This subject failed to achieve the 3 vs. 4 discrimination (47.59 ± 6.64%; *t*_9_ = 1.147, *p* = 0.281). The remaining four subjects did not achieve the 2 vs. 3 discrimination after performing, on average, 336.25 ± 110.07 trials.

To examine the influence of inter-trial interval (10 s vs. 10 min) we compared accuracy between Experiment 4A and Experiment 4B. In the 3 vs. 12 discrimination, we found no statistically evidence of an effect of time elongation (*t*_10_ = 0.285, *p* = 0.780) and the subjects of the two experiments were pooled for further analysis. Subjects’ accuracy was significantly greater than that expected at chance level (59.88 ± 6.95%; *t*_15_ = 5.689, *p* < 0.001). The GLMM revealed a significant improvement in the subjects’ accuracy over training sessions (χ^2^_1_ = 20.948, *p* < 0.001), no effect of condition (χ^2^_1_ = 0.030, *p* = 0.863). The session × condition interaction was significant (χ^2^_1_ = 6.844, *p* = 0.009), due to the fact that subject with the standard procedure (Experiment 4A) had a steeper learning curve than subjects which had a 10 min delay after each incorrect trial (Experiment 4B). The performance was significantly higher than that of Experiment 1 with automated device (GLMM; procedure: χ^2^_1_ = 18.474, *p* < 0.001; training session χ^2^_1_ = 29.835, *p* < 0.001; training session × procedure χ^2^_2_ = 8.775, *p* = 0.003), but did not reach the performance of manual training (χ^2^_1_ = 12.190, *p* < 0.001; session χ^2^_1_ = 25.220, *p* < 0.001; session × procedure χ^2^_1_ = 4.817, *p* = 0.028).

In the 2 vs. 3 discrimination, we found no statistically evidence of an effect of time elongation (*t*_4.455_ = 0.747, *p* = 0.493) and the subjects of the two experiments were pooled for further analysis. Accuracy was not greater than that expected at chance level (54.48 ± 10.03%; *t*_11_ = 1.546, *p* = 0.150). When the last six training sessions were considered, subjects’ accuracy was not greater than that expected at chance level (53.28 ± 9.62%; *t*_11_ = 1.179, *p* = 0.263).

### 2.7. Experiment 5: Reducing Cognitive Load

The automated procedure may increase the cognitive load of the task, thereby reducing fish learning performance, by requiring the fish to first attain a set of other skills (i.e., learn how to swim through the tank sectors, to approach the stimuli to cause food delivery, swim to the back chamber to prepare a new trial, etc.). In the present experiment, before subjecting guppies to the numerical task, we allowed them to become acquainted with these additional skills by giving them a series of three different non-numerical (shape, color, and size) discrimination tasks.

#### 2.7.1. Subjects, Apparatus, and Procedure

For this experiment, we used eight adult females. The conditioning chamber was identical to those used in Experiment 1. Subjects initially underwent a series of three visual discriminations: the first task required fish to discriminate between a circle (diameter 3 cm) and a 3 × 3 cm triangle on a white background; the second task was a color discrimination between red (red-green-blue color model: 255, 0, 0) and green (RGB: 0, 255, 0); and the third task was a size discrimination between a 3 × 3 cm square and a 1.5 × 1.5 cm square on a white background (ratio: 0.25). Fish that successfully completed this sequence underwent a fourth discrimination that was the same numerical discrimination 3 vs. 12 presented first in Experiment 1. We originally planned to continue the experiment by increasing the difficulty of the numerical task, as in previous experiments. However, as subjects appeared less motivated to complete the task, we decided to skip the last part the experiment and perform instead a second shape discrimination (different from the first one) to compare learning at the beginning of the experiment and after two months. The stimuli were a cross (3 cm in length and 0.75 cm in width) and a 3 × 1.5 cm horizontal bar. The learning criteria and other details of the procedure were identical to those adopted in Experiment 1. Thirteen subjects did not pass the pre-training phase and were replaced with new subjects.

We followed the same statistical approach described in previous experiments to analyze subjects’ accuracy. We additionally compared subjects’ performance between the first shape discrimination and the second shape discrimination using a GLMM. The model was fitted with training session and type of discrimination as a fixed factor, and the subject as a random factor.

#### 2.7.2. Results

Five out of eight subjects achieved the shape discrimination according to the primary criterion after performing on average 225.60 ± 156.19 trials (average number of days: 3.20 ± 2.17), whereas the remaining subject did not achieve the first discrimination (average number of trials: 435.67 ± 94.94; Table 7). Overall, the group of subjects did not show an accuracy greater than that expected at chance level (62.43 ± 15.88%; *t*_7_ =2.215, *p* = 0.062). Table 7 shows the individual performance with statistics for all subjects. The GLMM did not reveal a significant improvement in the subjects’ accuracy over training sessions (χ^2^_1_ = 0.370, *p* = 0.543).

All five subjects achieved the color discrimination according to the primary criterion after performing on average 416.00 ± 266.64 trials (average number of days: 5.60 ± 3.78). Overall, subjects’ accuracy was significantly greater than that expected at chance level (68.52 ± 13.29%; *t*_7_ = 3.115, *p* = 0.004). The GLMM revealed a significant improvement in the subjects’ accuracy over training sessions (χ^2^_1_ = 43.416, *p* < 0.001).

These 5 subjects also achieved the size discrimination according to the primary criterion after performing on average 373.20 ± 251.21 trials (average number of days: 4.60 ± 3.21). Overall subjects’ accuracy was significantly greater than expected by chance level (75.32 ± 10.15%; *t*_4_ = 5.580, *p* = 0.005, Table 7). The GLMM revealed a significant improvement in the subjects’ accuracy over training sessions (χ^2^_1_ = 10.594, *p* < 0.001).

Two out of five subjects achieved the 3 vs. 12 numerical discrimination according to the primary criterion after performing on average 564.00 ± 137.18 trials (average number of days: 7.5 ± 2.12), and two additional subjects reached the second learning criterion after performing on average 795.00 ± 7.07 trials. The remaining subject did not achieve it after 596 trials. Overall, subjects’ accuracy was significantly greater than that expected at chance level (60.29 ± 4.80%; *t*_4_ = 4.790, *p* = 0.009, Table 7). The GLMM revealed a significant improvement in the subjects’ accuracy over training sessions (χ^2^_1_ = 30.408, *p* < 0.001). The performance on this discrimination was significantly higher than that found in Experiment 1 with automated device (GLMM; procedure: χ^2^_1_ = 18.474, *p* < 0.001; training session: χ^2^_1_ = 29.835, *p* < 0.001; training session × procedure: χ^2^_1_ = 8.775, *p* = 0.003), but did not reach the performance of manual training (procedure: χ^2^_1_ = 12.190, *p* < 0.001; session: χ^2^_1_ = 25.220, *p* < 0.001; session × procedure χ^2^_1_ = 4.817, *p* = 0.028).

At this point in the experiment, one subject died, and the others did not appear very motivated to complete the test. We decided to subject the four remaining guppies to a new shape discrimination to determine whether, after nearly two months of experiments and more than two thousand trials, the performance of the subjects had declined. Only one out of four subjects achieved the second shape discrimination according to the primary criterion after 640 trials in 8 days, whereas the remaining subject did not (average number of trials: 746.67 ± 94.94). Subjects’ accuracy was not greater than that expected at chance level (60.20 ± 6.89%; *t*_3_ = 2.961, *p* = 0.060, Table 7). The GLMM did not reveal a significant improvement in the subjects’ accuracy over training sessions (χ^2^_1_ = 0.800, *p* = 0.371). A comparison of this task and the shape discrimination at the beginning of this experiment revealed a significant effect of the treatment (GLMM, task: χ^2^_1_= 10.187, *p* = 0.001), indicated that subjects showed a higher performance in the first shape discrimination task (Figure 2). There was no significant effect of the session (χ^2^_1_ = 0.016, *p* = 0.899), nor the interaction (χ^2^_1_ = 1.624, *p* = 0.203).

## 3. Discussion

In recent years, there have been numerous attempts to produce automated operant conditioning devices for small teleosts. A comparison of studies employing traditional training methods and studies employing automated devices seems to suggest that the latter enhance performance on some tasks [21,28] but worsen it on others [27,30,32]. Using a direct comparison of the two methods, we demonstrated that, when all the parameters of the tasks are controlled for, automated methods reduce the subjects’ performance on a numerical discrimination task. However, we showed that residency and reduction of cognitive load moderately improved the subjects’ performance.

### 3.1. Comparison of Manual and Automated Training

In Experiment 1, we compared in the same numerical task subjects trained with a recently developed automated device [28,30] and subjects trained with a traditional approach, a manual operant conditioning protocol that was uses in many previous studies [31,42,43]. We used the protocol that was previously found to be the most efficient in training fish on a numerical discrimination [23]. The subjects were first tested on an easy numerical contrast and the difficulty of the task progressively increased until the fish failed to reach the criterion. This protocol is ideal for making comparisons between different conditions because it allows comparing both learning performance within a single numerical task and the numerical acuity estimated by administering task of increasing difficulty. All of the subjects manually trained achieved the easiest discriminations, 3 vs. 12 and 2 vs. 3 items. Additionally, the two following, more difficult, discriminations were achieved by several subjects (3 vs. 4: 9 out of 11 subjects; 4 vs. 5: 3 out of 8 subjects). The 5 vs. 6 discrimination was not achieved by any subject. This result completely overlaps that obtained in the previous study with a different manual training protocol [23] and could not be attributed to an observer bias effect, as showed by a very high interrater reliability of scoring method. Conversely, no subject trained with automated device reached the criterion in 3 vs. 12 discrimination and only 3 out of 8 fish had a percentage of correct responses significantly above chance. It is worth noting that, with traditional training procedure, guppies were allowed a maximum of 120 trials for each discrimination, and, in most cases, they reached the criterion after a few dozen trials, whereas with the automated device they were allowed to perform up to a thousand trials. The poor numerical performance with automated training procedures observed in a previous study [30] is, thus, confirmed even with the adoption of a most efficient protocol, which starts from simple discriminations to gradually increase the difficulty. We can exclude that the low performance in the Skinner box is merely due to the inadequacy of the apparatus or the procedure since with the same equipment and procedure guppies rapidly reach up to 90% correct response in other types of discrimination, such as color and shape discrimination [28]. This performance is actually higher than previously reported in fish and also superior to that of many warm-blooded vertebrates. Since the proposed numerical tasks are easily discriminated by guppies trained with various non-automated protocols ([23,31], the present study), the cause of the poor performance must be sought in one or more characteristics that differentiate the two approaches to training.

One important difference regards the time available to the subjects to gather information about the stimuli, analyze them and make a decision. Indeed, a trade-off between decision speed and accuracy was shown for a variety of species [48,49,50]. Skinner box are typically very reduced in size compared with manual training apparatuses [21,27,51,52]. Specifically, in our study, the distance between the exit from the corridor and the line of choice was 22 cm for the manual training and only 6 cm (approx. one body length) in the automated training. This resulted in a choice time on average three times longer for the manual procedure. The short time interval (~1 s) of the Skinner box may be enough for perceiving the color or the shape of two objects but not sufficient to make an accurate numerical discrimination. In a numerical discrimination, stimuli change for every trial and the subject must independently estimate the number of items in the two stimuli, compare these two quantities and thereby decide based on a learned rule. In addition, if number discrimination requires a longer processing time, the possibility of inhibiting and correcting initially wrong choices may be very reduced in the Skinner box due to the much shorter time allowed to the subject before it reaches the choice areas [30].

An additional difference between color or shape discriminations and numerical discriminations is that the latter requires the fish to observe both stimuli in their entirety. A subject, for example, can perform a correct color discrimination even if one or both stimuli are partly hidden from view. Conversely, a larger of two numerosities may become the smaller one if some of its items are not visible. The tiny space of the automated conditioning chamber may cause a problem in visibility of the stimuli that was not present in the large tank used for manual training.

Another possibility for explaining the different performances is that automated procedures increase the cognitive load of the task by simultaneously requiring that the fish learn to solve the discrimination, to swim fluidly through the sectors, to associate its approach to the stimuli with food delivery, and to swim back to the starting chamber to launch a new trial. This hypothesis is corroborated by the observation that in experiments with automated training procedures many subjects are discarded before the training phase starts due to their inability to learn how to operate the device ([28,32,53], the present study). In many vertebrates, including fish, the addition of concurrent tasks increases cognitive load and decreases the performance on the primary task [54,55]. This is expected to occur, particularly when subjects are performing complex tasks [36,37]. Although it is not clear whether numerical discriminations are more cognitively demanding for fish than color or shape discriminations [56], they certainly imply a larger number of cognitive operations. In a shape or color discrimination, a subject sees the same pair of stimuli throughout the experiment and can solve the task simply by learning to approach the positive stimulus or move away from the negative one without necessarily making a comparison of the two. Conversely, in a numerical discrimination task, the two stimuli change at each trial because position, size and density of the items vary systematically. In each trial, the subject must make a new estimate of the two quantities, compare the two numerosities and, hence, apply the learned rule (e.g., choose the larger one). It is plausible that the additional cognitive load related to the use of the automated device is particularly detrimental when a fish is required to perform such a task.

Finally, the two procedures differ in the amount of manipulation received by the subjects during the experiment. As usual for conditioning protocols of fish (e.g., [43,57,58]), in the manual training experiment, guppies resided for the whole experiment in their experimental tank. In the automated training experiment, by contrast, they resided for most of the time in their home tank, and once or twice a day, they were netted and transferred to the conditioning chamber for the duration of the test. Netting and change in the environment, even for a short period, may cause severe and long-term stress in teleosts [59,60,61,62], and even chronic stress in case of repeated events [63]. Given the well-known negative impact of stress on cognitive performance [39,40,41], the manipulation required for the automated procedure may cause fish reduced discrimination learning success. Being in a new environment is also expected to cause short term increase in vigilance against predators, which typically reduces the attention devoted to foraging activities and decreases food-finding efficiency [54,55,64].

Other differences between the two methods of training could potentially be involved. For example, the number of trials per sessions or the presence of correction trials have been sometimes found to affect learning rate in other organisms [65,66]. However, they do not normally have dramatic effects on task achievement. In addition, an automated procedure with all these features led to performance equal or even superior to the traditional methods with color or shape discrimination [28] and with the discrimination of the size of stimulus (see Experiment 5) and it is difficult to devise hypotheses that easily explain why these features should selectively compromise learning in numerical tasks.

### 3.2. Can Automated Training Devices Be Improved?

Following the above hypotheses, in the second part of this study we tested four modified versions of the setup to see whether we could detect the key factors and reduce the gap in numerical discrimination performance between the automated training approach and the other methods.

*Elongation of decision time*. The first variant was intended to force guppies to take a longer time interval before making their choice (Experiment 2). The size of the conditioning chamber was increased, almost doubling the distance between the corridor and choice areas, and a transparent barrier was placed between the corridor and the choice areas to further lengthen the distance to reach the stimuli. Only one out of the six subjects learned to discriminate the 3 vs. 12 numerical discrimination, but this subject failed the subsequent 2 vs. 3 discrimination. These results suggest that the lengthening of the decision time, at least as it was induced here, is not enough to improve performance.

*Removal of internal partitions.* In Experiment 3, we further modified the conditioning chamber of the previous experiment by removing all internal dividers so that the animal had a complete view of the two stimuli from afar, as happened in the manual training tanks. None of the ten subjects tested in this experiment learned the 3 vs. 12 discrimination, suggesting that the visibility of the stimuli was probably not a crucial factor in determining the difference between the two training approaches of Experiment 1.

*Subjects resident in the conditioning chamber*. In Experiment 4, we modified the chamber so that we could keep the experimental subject there for the entire duration of the experiment, as was the case in manual training experiment. With this modification, 12 out of the 16 subjects achieved the 3 vs. 12 numerical discrimination, a considerable improvement on the setup of Experiment 1. Fish likely adapt less well to being moved daily between the home tank and the experimental chamber, relative to domestic species of mammals and birds, such as rats and pigeons for which the original Skinner box was developed. In the rainbowfish, *Melanotaenia duboulayi*, familiarity with the testing apparatus was found to increases task performance probably because familiarity decreases stress, allowing subjects to pay more attention to the task [67]. This hypothesis may also explain the unexpectedly low performance in numerical tasks found in a study in which guppies undergo a traditional manual training but were moved in turn to the conditioning chamber for the trial [44].

In order to allow the subject to reside in the apparatus, in this experiment we had to increase the size of the conditioning chamber. Therefore, in this experiment the size of the chamber is a potential confounding factor. One might for example argue that guppies are not at ease in tiny spaces and would be stressed by being tested in small apparatuses. However, in their natural habitat, guppies avoid open spaces and are generally found in the proximity of the river’s margins, in small pools that contain a fraction of the volume of our Skinner box and they often forage in small gaps of thick vegetation that can host just one fish [68,69,70]. As a further confirmation of the fact that the size of the chambers per se is unlikely to represents a key factor, excellent performance was obtained with the smallest version of the conditioning chamber in other tasks [28], and some studies have found good numerical performances using very small manual training apparatuses (e.g., [71]). It is worth noting the global performance of fish resident in the conditioning chamber was still inferior to the manually trained group of Experiment 1. Being resident in the test apparatus is not the only factor that determines the difference between the two approaches of training used here.

One of the advantages of the Skinner box was that a single small-sized equipment could be used to train several rats or pigeons in rotation. Maintaining the fish in the conditioning chamber improves performance but suffers the major drawback that it implies the use of a large tank for each experimental subject and therefore experiments require much more space and equipment. A compromise may be the one adopted in a previous study [30], in which the subjects are semi-residential, i.e., they live in a tank that houses the conditioning chamber to which they are moved in turn at the time of the test. In this way, the relocation is rapid and the subject stays in identical physical, chemical and olfactory conditions for the duration of the experiment. A rough comparison between the results of the two studies suggests that this solution is sufficient to achieve some improvement. In our Experiment 1, only 3 of the 8 subjects had a performance greater than that expected by chance; this happened for all subjects but one in the previous study, though in the latter, the ratios to discriminate were more difficult. Indeed, the performance significantly differed between the two studies (Mann–Whitney U test; *p* = 0.045). It is also possible to devise a conditioning chamber in which the subjects are accustomed to leave their group in turn, and swim to the testing area at the time of test, a common practice in primate studies (e.g., [72,73,74]).

The results of Experiment 4 have potential implications for the welfare of the laboratory studies in fish. Moving repeatedly the subject forth and back between home cage and the testing apparatus is a common practice of cognition studied in fish, as it is in mammals and birds (e.g., [4,18,75]). However, this practice could induce stress in fish more than in other vertebrates, especially if the experiment is prolonged, as also suggested by the outcome of Experiment 5 (see below).

In this experiment we also manipulated inter-trial interval, another factor that frequently differentiates automated conditioning approach from traditional ones and that also differed considerably in Experiment 1. We increased the inter-trial interval in half of the subjects from 10 s to 10 min, making it very similar to the inter-trial interval adopted in the manual training experiment, but we found no evidence that this treatment affected discrimination learning or numerical acuity of the subjects.

*Reducing cognitive load.* The aim of Experiment 5 was to reduce the cognitive load during the execution of the task by uncoupling learning the use of the conditioning chamber from learning the numerical discrimination. Before being admitted to the numerical discriminations, guppies underwent a shape, a color, and a size discrimination. The performance on these three tasks was substantially similar to that observed in previous studies employing different training procedures [23,29]. When subjected to the 3 vs. 12 numerical discrimination, guppies showed an evident improvement compared to the automated training of Experiment 1, although their performance was significantly lower than in the experiment with manual training.

At the end of this series of tasks, after nearly two months of experiments and more than two thousand trials, one animal died, and others showed signs of stress. Therefore, we decided to stop the experiment and we do not know the quality of the performance of the fish of this treatment on the more challenging numerical discriminations. This outcome tells us that, unlike monkeys and mice, fish cannot be tested for unlimited time durations, at least not if it is necessary to move them frequently from their home tank to an experimental chamber. A solution to the problem of moving fish may be to have the subject reside in the experimental chamber as in Experiment 4, in which this solution seems to have brought benefits in terms of performance. It is necessary to verify whether guppies that are resident or semi-resident in the conditioning chamber can do a thousand trials without suffering or diminishing their performance.

The first part of Experiment 5 confirms previous finding about the efficiency of our automated training device for discrimination learning [28,32]. Considering the first days of training in each discrimination, before the best performing subjects were admitted to the next discrimination, guppies evidenced an excellent performance in color, shape and size discriminations reaching 75–80% correct choices in few sessions. It is interesting to note that the first part of this experiment also shows that guppies, like mammals and birds [76,77], can learn a series of different discriminations in rapid sequence without apparent interference of one task on the subsequent.

### 3.3. A Comparison with Other Studies Using Automated Conditioning Devices

As mentioned in the Introduction, learning, memory, and many other cognitive functions are investigated in warm-blooded vertebrates using primarily automated procedures whereas manual training is the standard procedure used with teleost fish, with other lower vertebrates and with invertebrates. Some laboratories have attempted overcome this limitation by developing a Skinner-box apparatus for fish. In the 1960s and 1970s, many studies investigated learning and memory in the goldfish using fully automated operant conditioning devices with the aim to compare a fish species with the classical mammalian and avian model species [78,79,80]. When discrimination learning experiments were performed, goldfish appeared to master well color [79,80,81] as well as shape discrimination [82,83].

In recent years, some small tropical fish, in particular zebrafish, medaka and guppy, have become important models in neurobiological research. Taking advantage of the emerging digital technologies, there have been half a dozen independent attempts to implement automated devices, obtaining mixed results. Manabe and colleagues [18] set up a computer-controlled operant apparatus for small fish, in which stimuli were presented by means of LEDs and the approach to response key was sensed through an optical fiber sensor, which triggered an automated feeder delivering small amount of food. With the same apparatus, Kuroda and colleagues [19], using a color discrimination, demonstrated that zebrafish can learn a reversal learning task.

Color discrimination is, by far, the most frequently investigated discrimination task in the remaining studies that employed automated training approach. Another device allowed for the simultaneous training of six subjects, each resident in an adjacent chamber. A computer tracked the position of each fish, controlled stimulus presentation on a LCD monitor and commanded the delivery of the food reward [21]. The apparatus was tested by training zebrafish on a blue-green color discrimination. Subjects slowly improved their performance over 30 daily sessions of 20 trials each, reaching 80% of correct choices at the end of experiment. Conversely, Miletto Petrazzini and colleagues [27], who tested zebrafish with a commercially available operant conditioning apparatus, found a much lower performance on two color discriminations. With the exception of the latter study, the others were in agreement with the results obtained with our device in the present study or in previous ones [28,32].

Two studies conducted in our laboratory investigated shape discrimination. Guppies showed good learning performance, fully comparable with the results of manual training experiments [28]. Conversely, with zebrafish, no subject reached the criterion on the same shape discrimination task, even after extended training [32], despite this species showing 75% accuracy in color discrimination tasks with the same device. There is scarce information about the discriminative abilities of this species except for color discrimination. Zebrafish proved able to discriminate novel shapes from familiar ones [34,35]. However, the sole study that examined shape discrimination in zebrafish using appetitive conditioning was an interspecific study in which zebrafish showed a rather poor performance compared to other teleosts [25].

To date, only another laboratory investigated numerical discrimination in a fish, *Gambusia affinis*, using an automated system [33]. The device and the procedure differed from those used in the other studies. During the training phase, the quantities to discriminate were presented on two monitors placed at the opposite ends of a rectangular tank and a computer-operated system delivered a food reward when the subject approached the correct stimulus. Learning was measured in a probe trial in which the subject was exposed to the stimuli without reinforcement, and the percentage of time spent near the positive stimulus was calculated from video-recordings. This experiment replicated a study on the same species with an identical procedure, but in which the training phase was done manually [84]. A rough comparison of the two studies shows that, even in this case, the performance on numerical discrimination tasks was better when fish were trained manually (*p* = 0.016; Welch *t*-test).

## 4. Conclusions

In summary, the results of this study confirm that the automated training device we developed modelling the classical Skinner boxes can satisfactorily be used to train guppies in some tasks (i.e., color, shape, and size discriminations) but are totally inadequate for other tasks, such as a numerical discrimination. Similar inconstancy in performance have been reported with other automated devices in different fish species and with different tasks which suggests that small laboratory teleosts may be limited in their capacity to cope with some undetermined aspects of the automated approach to training. These limitations need to be overcome, as they presently prevent an easy and straightforward comparison of teleosts with the other vertebrates.

In a series of experiments, we introduced modifications to the automated training apparatus and procedure in an attempt to fill the performance gap with the best performing manual training procedure available for guppies. The difference did not appear to be related to a longer decision time in manual procedure nor to the different visibility of the stimuli in the two procedures. Being resident in the test tank improved the performance of the subjects but this factor alone did not lead to the same performance of the manual training experiment. A similar effect was obtained by reducing the cognitive load through the temporal dissociation of the familiarization with the functioning of the automatic equipment from the numerical discrimination task.

It, therefore, seems that there is no single factor that explains the different efficiency of the two procedures tested in this study, but rather several factors acting synergistically to determine the different performance. There are many other differences between the two methods compared in this study, and in future, it will be necessary to investigate other factors that may be important. One interesting difference that was not addressed concerns the way the stimuli are presented in the two approaches, generated on a computer screen or stuck onto real objects introduced into the water. Computer-generated stimuli have already been used successfully in fish for visual discrimination tasks [28,85,86] and a vast literature shows that fish react to stimuli presented on the monitor as they react to the real objects [87,88,89,90,91]. However, it is possible that stimuli introduced into the tank are much more salient and focus the subject’s attention on the task. Indeed, in several species the salience of the stimulus was shown to influence the performance on discrimination tasks (capuchin monkeys [92]; pigeons [93]; keas [94]) and there is also partial support for this effect in the guppy [46]. It will be interesting to verify whether an automated procedure that commands the presentation of solid stimuli introduced into the subject’s tank allows for a performance comparable to that we obtained in our study with the reference manual training procedure.

## Figures and Tables

**Figure 1 animals-11-01397-f001:**
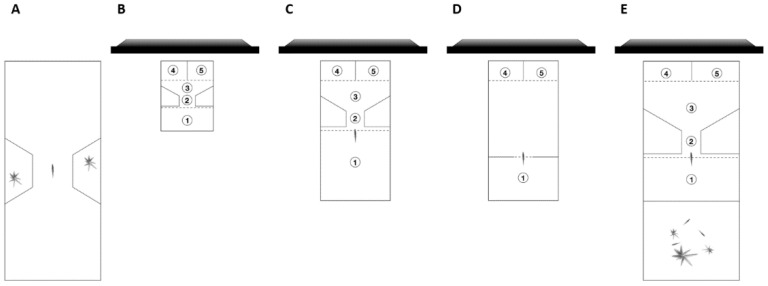
Aerial view of the (**A**) manual training apparatus and (**B**–**E**) the four versions of automated conditioning chambers used in this study. Each chamber was internally divided into a (1) starting area, a (2) a corridor, (3) V-shaped decision area, and (4,5) two choice areas. An LCD computer monitor projected the stimulus in each choice areas. (**B**) Automated conditioning chamber used in Experiment 1 and 5. (**C**) Automated conditioning chamber used in Experiment 2; the chamber was inserted into a larger tank (20 × 50 × 32 cm) provided with a gravel bottom, natural vegetation housing immature companions. (**D**) Automated conditioning chambers used in Experiment 3. (**E**) Automated conditioning chambers used in Experiment 4A and 4B.

**Figure 2 animals-11-01397-f002:**
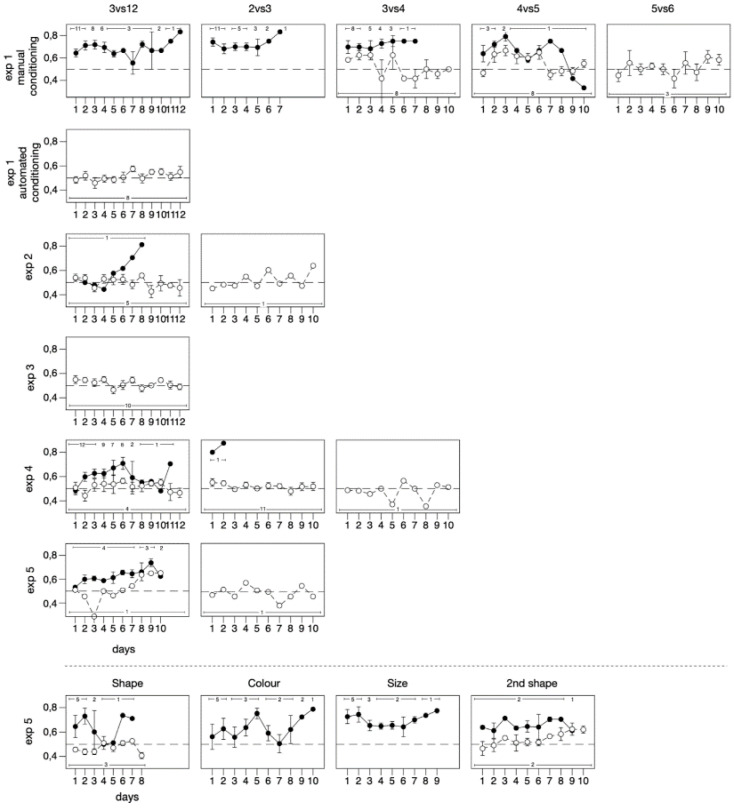
Performance of guppies in manual and automated experiments divided for the task. Lines with black dots represented the performance (mean ± standard error) of subjects that achieved the discrimination according to the learning criteria. Performance of subjects in Experiments 4A and 4B were pooled. Numbers in the upper part correspond to the sample size of subjects tested in each training session. This number decreased as the subjects reached criterion and were admitted to the subsequent discrimination. Lines with white dots represented the performance of subjects that failed to achieve the discrimination (sample size in the lower part). The dotted line represents chance performance (50% correct responses).

**Table 1 animals-11-01397-t001:** Individual performance of guppies in Experiment 1 with manual conditioning. In each cell, we reported the percentage of correct responses (mean standard deviation), the number of correct responses/number of total choices, and the *p* value calculated with the binomial test for each task. Subject N10 stopped participating after 72 trials of the 2 vs. 3 discrimination. Asterisks indicate the subjects that chose the correct stimulus more often than expected by chance. n/a is reported when a subject did not perform the task.

Experiment 1: Manual Conditioning					
Subjects	3 vs. 12	2 vs. 3	3 vs. 4	4 vs. 5	5 vs. 6
1	87.50 ± 5.89%; 21/24	79.17 ± 5.89%; 19/24	75.00 ± 11.79%; 36/48	59.17 ± 11.42%; 71/120	n/a
	*p* < 0.001 *	*p* = 0.007 *	*p* < 0.001 *	*p* = 0.055	
2	70.83 ± 4.45%; 68/96	75.00 ± 23.57%; 18/24	75.00 ± 3.57%; 18/24	56.67 ± 14.59%; 68/120	n/a
	*p* < 0.001 *	*p* = 0.023 *	*p* = 0.023 *	*p* = 0.171	
3	66.67 ± 11.79%; 32/48	66.67 ± 9.62%; 32/48	65.00 ± 25.91%; 39/60	56.67 ± 13.49%; 68/120	n/a
	*p* = 0.029 *	*p* = 0.029 *	*p* = 0.027 *	*p* = 0.171	
4	64.17 ± 7.91%; 77/120	79.17 ± 5.89%; 19/24	53.70 ± 10.54%; 64/120	n/a	n/a
	*p* = 0.002 *	*p* = 0.007 *	*p* = 0.523		
5	68.75 ± 14.23%; 33/48	75.00 ± 0.00%; 18/24	67.86 ± 8.91%; 57/84	69.44 ± 17.35%; 25/36	50.83 ± 10.72%; 61/120
	*p* = 0.013 *	*p* = 0.023 *	*p* = 0.001 *	*p* = 0.029 *	*p* = 0.927
6	75.00 ± 11.79%; 18/24	91.67 ± 11.79%; 22/24	75.00 ± 11.79%; 18/24	61.67 ± 13.72%; 74/120	51.67 ± 11.65%; 62/120
	*p* = 0.023 *	*p* < 0.001 *	*p* = 0.023 *	*p* = 0.013 *	*p* = 0.784
7	66.67 ± 14.43%; 24/36	65.00 ± 13.69%; 39/60	73.33 ± 6.97%; 44/60	55.00 ± 11.91%; 64/120	n/a
	*p* = 0.065	*p* = 0.027 *	*p* < 0.001 *	*p* = 0.523	
8	66.67 ± 13.61%; 32/48	70.83 ± 8.33%; 34/48	50.00 ± 13.61%; 60/120	n/a	n/a
	*p* = 0.029 *	*p* = 0.006 *	*p* = 1.000		
9	62.04 ± 13.89%; 67/108	75.00 ± 11.79%; 18/24	66.67 ± 16.67%; 24/36	52.50 ± 13.64%; 63/120	n/a
	*p* = 0.016 *	*p* = 0.023 *	*p* = 0.065	*p* = 0.648	
10	75.00 ± 23.57%; 18/24	69.44 ± 4.30%; 50/72	n/a	n/a	n/a
	*p* = 0.023 *	*p* = 0.001 *			
11	69.44 ± 12.73%; 25/36	66.67 ± 11.79%; 56/84	79.17 ± 5.89%; 19/24	75.00 ± 0.00%; 18/24	52.50 ± 13.64%; 61/120
	*p* = 0.029 *	*p* = 0.003 *	*p* = 0.007 *	*p* = 0.023 *	*p* = 0.927

**Table 2 animals-11-01397-t002:** Individual performance of guppies in Experiment 1 with automated conditioning. Asterisks indicate the subjects that chose the correct stimulus more often than expected by chance.

Experiment 1: Automated Conditioning		
Subjects	3 vs. 12	3 vs. 12 (Last 6 Sessions)
1	50.52 ± 7.19%;	51.07 ± 8.04%;
	268/523;	150/289;
	*p* = 0.600	*p* = 0.556
2	57.01 ± 10.67%;	60.51 ± 9.42%;
	355/621;	134/223;
	*p* < 0.001 *	*p* = 0.003 *
3	54.76 ± 6.22%;	56.59 ± 7.20%;
	362/662;	209/373;
	*p* = 0.018 *	*p* = 0.038 *
4	55.82 ± 6.88%;	59.42 ± 6.43%;
	475/839;	269/448;
	*p* < 0.001 *	*p* < 0.001 *
5	46.88 ± 11.78%;	50.71 ± 7.48%;
	151/324;	74/143;
	*p* = 0.243	*p* = 0.738
6	53.04 ± 6.04%;	54.58 ± 6.72%;
	301/570;	163/297;
	*p* = 0.191	*p* = 0.104
7	44.69 ± 10.80%;	44.69 ± 12.31%;
	139/301;	60/127;
	*p* = 0.205	*p* = 0.595
8	49.64 ± 9.58%;	50.43 ± 8.67%;
	209/423;	87/164;
	*p* = 0.846	*p* = 0.482

**Table 3 animals-11-01397-t003:** Individual performance of guppies in Experiment 2. Asterisks indicate the subjects that chose the correct stimulus more often than expected by chance. n/a is reported when a subject did not perform the task.

Experiment 2		
Subjects	3 vs. 12	2 vs. 3
1	58.39 ± 12.40%;	49.06 ± 5.30%;
	219/339;	338/650;
	*p* < 0.001 *	*p* = 0.327
2	51.68 ± 7.64%;	n/a
	170/328;	
	*p* = 0.544	
3	53.70 ± 5.77%;	n/a
	368/679;	
	*p* = 0.032 *	
4	51.10 ± 8.10%;	n/a
	385/745;	
	*p* = 0.379	
5	45.83 ± 12.61%;	n/a
	142/288;	
	*p* = 0.860	
6	48.11 ± 9.92%;	n/a
	230/470;	
	*p* = 0.678	

**Table 4 animals-11-01397-t004:** Individual performance of guppies in Experiment 3. Asterisks indicate the subjects that chose the correct stimulus more often than expected by chance.

Experiment 3			
	Condition A		Condition B
Subjects	3 vs. 12	Subjects	3 vs. 12
1	53.35 ± 8.02%; 267/494;	1	54.14 ± 9.84%; 131/240;
	*p* = 0.079		*p* = 0.175
2	49.96 ± 7.37%; 179/352;	2	49.37 ± 10.74%; 172/347;
	*p* = 0.790		*p* = 0.915
3	53.82 ± 8.14%; 195/363;	3	47.43 ± 9.14%; 213/444;
	*p* =0.172		*p* = 0.420
4	52.67 ± 6.47%; 219/411;	4	53.03 ± 11.37%; 207/388;
	*p* = 0.200		*p* = 0.204
5	51.40 ± 7.17%; 253/485;	5	51.90 ± 8.14%; 200/376;
	*p* =0.364		*p* = 0.236

**Table 5 animals-11-01397-t005:** Individual performance of guppies in Experiment 4A. Asterisks indicate the subjects that chose the correct stimulus more often than expected by chance. n/a is reported when a subject did not perform the task.

Experiment 4A		
Subjects	3 vs. 12	2 vs. 3
1	61.38 ± 18.04%; 155/239;	54.92 ± 8.90%; 216/397;
	*p* < 0.001	*p* = 0.088
2	58.85 ± 13.62%; 246/395;	54.26 ± 9.20%; 297/532;
	*p* < 0.001	*p* = 0.008
3	55.50 ± 6.81%; 330/589;	n/a
	*p* = 0.004	
4	67.67 ± 11.94%; 125/181;	55.90 ± 5.74%; 319/559;
	*p* < 0.001	*p* < 0.001
5	56.64 ± 21.96%; 191/283;	55.00 ± 6.50%; 379/697;
	*p* < 0.001	*p* < 0.001
6	62.23 ± 12.46%; 239/382;	52.10 ± 7.33%; 202/391;
	*p* < 0.001	*p* = 0.544
7	53.92 ± 13.07%; 224/416;	43.70 ± 7.93%; 147/334;
	*p* = 0.128	*p* = 0.033
8	66.94 ± 4.60%; 196/292;	50.28 ± 8.02%; 316/621;
	*p* < 0.001	*p* = 0.688

**Table 6 animals-11-01397-t006:** Individual performance of guppies in Experiment 4B. Asterisks indicate the subjects that chose the correct stimulus more often than expected by chance. n/a is reported when a subject did not perform the task.

Experiment 4B		
Subjects	3 vs. 12	2 vs. 3
1	71.68 ± 9.04%; 127/175;	83.75 ± 5.30%; 134/160;
	*p* < 0.001	*p* < 0.001
2	60.32 ± 14.09%; 138/214;	53.44 ± 11.59%; 142/259;
	*p* < 0.001	*p* = 0.136
3	60.51 ± 13.89%; 148/212;	54.13 ± 5.03%; 207/382;
	*p* < 0.001	*P*= 0.113
4	46.00 ± 9.77%; 160/332;	n/a
	*p* = 0.546	
5	66.28 ± 14.73%; 136/196;	44.46 ± 7.78%; 114/250;
	*p* < 0.001	*p* = 0.184
6	48.78 ± 9.97%; 178/348;	n/a
	*p* = 0.708	
7	56.14 ± 6.63%; 195/339;	n/a
	*p* = 0.007	
8	65.25 ± 7.02%; 91/134;	51.79 ± 6.79%; 253/483;
	*p* < 0.001	*p* = 0.317

**Table 7 animals-11-01397-t007:** Individual performance of guppies in Experiment 5. Asterisks indicate the subjects that chose the correct stimulus more often than expected by chance. n/a is reported when a subject did not perform the task.

Experiment 5					
Subjects	Shape	Color	Size	3 vs. 12	2nd Shape
1	48.33 ± 7.66%; 190/374;	n/a	n/a	n/a	n/a
	*p* = 0.796				
2	54.20 ± 12.96%; 277/498;	60.66 ± 18.29%; 246/406;	64.58 ± 8.08%; 465/720;	60.91 ± 8.76%; 439/721;	64.06 ± 7.58%; 410/640;
	*p* = 0.014	*p* < 0.001	*p* < 0.001	*p* < 0.001	*p* < 0.001
3	65.64 ± 19.46%; 142/204;	88.13 ± 0.88%; 141/160;	73.77 ± 4.95%; 204/275;	62.19 ± 5.50%; 492/791;	56.38 ± 7.94%; 445/787;
	*p* < 0.001	*p* < 0.001	*p* < 0.001	*p* < 0.001	*p* < 0.001
4	75.88 ± 4.46%; 81/106;	59.06 ± 13.77%; 450/750;	67.01 ± 4.95%; 369/551;	52.03 ± 11.15%; 326/596;	n/a
	*p* < 0.001	*p* < 0.001	*p* < 0.001	*p* = 0.024	
5	46.96 ± 5.52%; 182/388;	n/a	n/a	n/a	n/a
	*p* = 0.243				
6	45.68 ± 3.80%; 251/545;	n/a	n/a	n/a	n/a
	*p* = 0.072				
7	81.88 ± 2.65%; 131/160;	76.54 ± 2.18%; 117/153;	83.13 ± 4.42%; 133/160;	61.82 ± 8.15%; 288/467;	52.66 ± 5.34%; 391/739;
	*p* < 0.001	*p* < 0.001	*p* < 0.001	*p* < 0.001	*p* = 0.122
8	81.88 ± 4.42%; 131/160;	58.20 ± 13.84%; 370/611;	88.13 ± 4.42%; 141/160;	64.50 ± 8.96%; 430/661;	67.68 ± 4.29%; 483/714;
	*p* < 0.001	*p* < 0.001	*p* < 0.001	*p* < 0.001	*p* < 0.001

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
