# Peer review of "Automated Operant Conditioning Devices for Fish. Do They Work?"

_animals, 2021, doi:10.3390/ani11051397_

Round 1

Reviewer 1 Report

  1. those obtained by other laboratories with similar devices, the limitations of

  2. 18  adopting automated devices for fish do not seem to compensate the benefits.

This is an important statement,  totally confirmed by their dta. And Authors (if possible) should enlarge their perspective to other automated behavioural tests, including eg rodent maze(s) and AI scoring of behavioural patterns. Maybe a couple of quotations in thei RDiscussion may be helpful.

  1. With

  2. 47  manual execution, many months and hundreds of hours of work are required to train each subject

  3. 48  [3-5]. The second advantage is that automated equipment allows for the control of every detail of the

  4. 49  experiment,

A few exemples may enlighten the readers, for the longest procedures.

  1. 57  tropical fish, in particular zebrafish (Danio rerio),

  2. Zebrafish is becoming a current substitute for a variety of rodent experiments. From one hand this is welcomed for ethical reasons. On the other, fish phylogenetic "distance" and "diversity" from human patients may well damage the validity of of animal models in general, if fish results are not valid when traslated to humans.  However, zebrafish here is enlisted among various fish species, yet its currrent role is very different and its contemporaly impact in biomedicine should deserve a few specifications.

  1. Finally, it is possible that automation frightens the subjects. While

  2. 105  rats and pigeons adapt very quickly to being manipulated and introduced into the Skinner

  3. 106  box periodically, fish may perceive a threat and therefore reduce the attention payed to the

  4. 107  task [26,27].

  1. 966  15. Manabe, K.; Dooling, R.J.; Takaku, S. An automated device for appetitive conditioning

  2. I wolder how much such a fish-adapted Skinner box is homologous to the bird and mammalian ones. A few specifications may help.

  1. 967  in zebrafish (Danio rerio). Zebrafish 2013, 10, 518-523.

  2. Again, zebrafish is a special case.

  1. In the training phase, each subject performed 12 trials per day for a maximum of 12

  2. 219  consecutive days for the 3 versus 12 numerical discrimination and 10 days for all other

If possible, an idea of the total duration (days\hrs) of the experimental phase may be added.

 750  Removal of internal partitions

  1. 751  In Experiment 3, we further modified the conditioning chamber of the previous

  2. 752  experiment by removing all internal dividers so that the animal had a complete view of the

  3. 753  two stimuli from afar, as happened in the manual training tanks.

Minor point. How this removal may frighten or disturb the animals under experimentation?

  1. 789  Reducing cognitive load

  2. 790  The aim of Experiment 5 was to reduce the cognitive load

 I do undertand that the term "cognitive load" gained consensus in the cognitive sciences literature. However for the readership of this Journal an even summarized definition of it may render the narration more fluid.

  1.  

.

Author Response

 RESPONSE TO REFEREE 1

## Thank you very much for your helpful comments. We largely rewrote the manuscript and considered your suggestions.

This is an important statement,  totally confirmed by their dta. And Authors (if possible) should enlarge their perspective to other automated behavioural tests, including eg rodent maze(s) and AI scoring of behavioural patterns. Maybe a couple of quotations in thei RDiscussion may be helpful.

and

I wolder how much such a fish-adapted Skinner box is homologous to the bird and mammalian ones. A few specifications may help.

## We now have better specified that our automated device was originally intended to be similar to the devices used to train rats and pigeons (Lines 104-115). We also reported more evidence of low performance in fish when using automated devices (Lines 87-93).

A few exemples may enlighten the readers, for the longest procedures.

## We provided some example to help the readers through the lecture (Lines 109-115; 116-121)

Zebrafish is becoming a current substitute for a variety of rodent experiments. From one hand this is welcomed for ethical reasons. On the other, fish phylogenetic "distance" and "diversity" from human patients may well damage the validity of of animal models in general, if fish results are not valid when traslated to humans.  However, zebrafish here is enlisted among various fish species, yet its currrent role is very different and its contemporaly impact in biomedicine should deserve a few specifications.

## We added some references to highlight the importance of small tropical fish in neurobiological research (Line 60).

If possible, an idea of the total duration (days\hrs) of the experimental phase may be added.

## We provided in Table 1-7 the number of trials each subject performed in each discrimination. We also provided the average number of days that fish required to achieve the numerical discrimination according to the primary criterion (see Results sections for each experiment).

Minor point. How this removal may frighten or disturb the animals under experimentation?

## We better explained that partition was removed before the experiment started (Lines 492-494).

I do undertand that the term "cognitive load" gained consensus in the cognitive sciences literature. However for the readership of this Journal an even summarized definition of it may render the narration more fluid.

## We provided a specification of what we intended as cognitive load (Lines 130-131).

Reviewer 2 Report

Automated SOPs for operant conditioning is a laudable goal and potentially a valuable contribution to behavioural biology. 

  1. The main goal of this study, as I understood it, was to compare guppy performance in cognitive tasks between manual conditioning and automated conditioning. However, the test of this question is hopelessly confounded by the dimensions and features of the test arena used for each type of conditioning. The manual conditioning used a relatively large test arena (Fig. 1 A), whereas automated conditioning put test subjects in a tiny box (Fig. 1 B-D). Differences in guppy performance between those in chamber A and those in chamber B-D may be explained entirely by stress induced from being confined in such a small space. Evidence for this stress is the high number of test subjects that failed to eat in the small test chambers. In Arena A (manual conditioning) 1/12 failed (8%), whereas failure rates in Arenas B, C and D were 13/21 (62%), 31/37 (84%), and 17/27 (63%), respectively. Test subjects were not likely to provide quality data and a valid test of the research question if they were too stressed to eat because response to a food reward was the variable used to determine performance on the cognitive tasks. An experimental test of the relative efficacy of manual versus automated conditioning requires a standard testing chamber for both methods, such as arena A. The present experimental design is confounded and the data are uninterpretable. 
  2. Experiment 5 on the effect of cognitive load has merit as a stand-alone question independent of the method of conditioning. Notably, the test arena was larger (Fig. 1 E), but even here 13/21 (62%) test subjects were removed in pre-training sessions because they refused to eat. Moreover, sample size was reduced from 8 to 7 when one died, and the remaining subjects showed signs of lacking motivation (lines 604, 643), calling into question what, if anything, readers are to take from data collected from 7 unmotivated guppies. 
  3. Experiment 2 is ostensibly about testing the effect of increasing decision time, but it is in fact about providing more space for the fish to move. Experiment 3 is billed as the effect of removing internal partitions, but it is in fact again about providing more space for the test subjects to move. Experiment 4 is defined as a test of residency in the test chamber, but the chamber is also significantly larger than Arenas B and C. There is no consistency to the experimental design.
  4. Word smithing:
    1. line 101: make a decision
    2. line 106: paid
    3. line 108: has been difficult to discern among these
    4. line 111: protocols
    5. line 124: tasks
    6. line 161: used
    7. line 175: used
    8. line 178: versions
    9. line 179: was
    10. line 229: using a binomial test
    11. line 233: the case
    12. line 246: area
    13. line 247: allowed projection of stimuli
    14. line 251: rods? (instead of "sticks"?)
    15. line 252: the servomotor
    16. line 258: conditions
    17. line 268: minute
    18. line 280: sessions
    19. line 319: P=0.05?
    20. line 369: delete 's'
    21. line 604: motivated

Author Response

RESPONSE TO REFEREE 2

Automated SOPs for operant conditioning is a laudable goal and potentially a valuable contribution to behavioural biology. 

  1. The main goal of this study, as I understood it, was to compare guppy performance in cognitive tasks between manual conditioning and automated conditioning. However, the test of this question is hopelessly confounded by the dimensions and features of the test arena used for each type of conditioning. The manual conditioning used a relatively large test arena (Fig. 1 A), whereas automated conditioning put test subjects in a tiny box (Fig. 1 B-D). Differences in guppy performance between those in chamber A and those in chamber B-D may be explained entirely by stress induced from being confined in such a small space. Evidence for this stress is the high number of test subjects that failed to eat in the small test chambers. In Arena A (manual conditioning) 1/12 failed (8%), whereas failure rates in Arenas B, C and D were 13/21 (62%), 31/37 (84%), and 17/27 (63%), respectively. Test subjects were not likely to provide quality data and a valid test of the research question if they were too stressed to eat because response to a food reward was the variable used to determine performance on the cognitive tasks. An experimental test of the relative efficacy of manual versus automated conditioning requires a standard testing chamber for both methods, such as arena A. The present experimental design is confounded and the data are uninterpretable. 
  2. Experiment 5 on the effect of cognitive load has merit as a stand-alone question independent of the method of conditioning. Notably, the test arena was larger (Fig. 1 E), but even here 13/21 (62%) test subjects were removed in pre-training sessions because they refused to eat. Moreover, sample size was reduced from 8 to 7 when one died, and the remaining subjects showed signs of lacking motivation (lines 604, 643), calling into question what, if anything, readers are to take from data collected from 7 unmotivated guppies. 
  3. Experiment 2 is ostensibly about testing the effect of increasing decision time, but it is in fact about providing more space for the fish to move. Experiment 3 is billed as the effect of removing internal partitions, but it is in fact again about providing more space for the test subjects to move. Experiment 4 is defined as a test of residency in the test chamber, but the chamber is also significantly larger than Arenas B and C. There is no consistency to the experimental design.

## Thank you very much for your helpful comments.

## You are right in saying that an experimental test of the relative efficacy of manual versus automated conditioning would require a standard testing chamber and procedure for both methods.

## As the Editor's suggestion, we rewrote the article, that is now no longer as a comparison between automatic training and manual training, but a comparison between two previously used methodologies that at least for some tasks seemed to yield very different outcomes. One is an automatic training approach similar to that used for rats or pigeons (e.g., small cage, high number of training per session, etc.) the other a traditional method for training fish (e.g., fish resident in testing apparatus, manual training, few daily trials, etc.; see Lines 104-125).

## The first experiment of the study aimed at verifying if the two procedures really differed in training effectiveness, using subjects of the same stock, age and sex and especially the same numerical task (Lines 146-150).

## Confirmed the difference, in the second part of the study, we modified the less efficient procedure in order to try to improve its effectiveness (Lines 126-145; 151-158). The differences between the two approaches are many and it would be difficult to study them all in the same study. Therefore, we have identified five factors that in our opinion were more likely to be responsible for the scarce effectiveness of one method in numerical tasks. Two of these factors were found to improve the performance of the subjects, three were influential.

## There are many other differences between the two approaches and in future, it will be necessary to investigate other factors that may be important. For example, we have not directly investigated the influence of the size of apparatus. In addition, as you noticed, in order to allow the subject to reside in the apparatus, we had to increase the size of the conditioning chamber. Therefore, the size of the chamber was a potential confounding factor in experiment four. However, various evidences suggest that it unlikely that the size per se represents the key factor.

## First, previous studies suggest that the chamber A (i.e., devices used in Experiment 1 and 5) is not too tiny or otherwise inadequate for operant conditioning. Using the same equipment and procedure, guppies rapidly reached up to 90% correct response with other types of discriminations (colour and shape; Lines 816-819; 863-869). This performance is much better than previously reported with any procedure in fish and better than the performance reported for many warm-blooded vertebrates.

## Second, in their natural habitat, guppies and their relatives are generally found in the proximity of the river's margins in small pools that contain a fraction of the volume of our skinner box and they often forage in small gaps of thick vegetation that can host just one fish (Lines 812-816).

## We also better explained that the larger number of fish discarded in the automated approach experiments was due to their inability to learn within few days how to operate the device and that the same was observed in other studies with automated device (in manual training paradigms, less fish are discarded as subjects are not required to learn how to operate the system and trials are initiated at intervals by the experimenter) (Lines 750-752).

## Finally, we carefully correct the manuscript following your minor suggestions. We are thankful for your efforts.

Reviewer 3 Report

Dear authors

I hope this message will find you well.

ID: animals-1064235

Title:   Automated operant conditioning procedures for fish. Do they work?

Authors: Elia Gatto *, Maria Santacà, Ilaria Verza , Marco Dadda and Angelo Bisazza

General comments:

The paper by E. Gatto and colleagues is undoubtedly a very good and interesting paper. A scrupulous description of methods and a large collection of information which are reported and expressed in an excellent way. I have only few suggestions.

-Please format the manuscript based on the author's guide. Methods together and all results together can help the readers.

-Is there a reason you only used female specimens?

-Lines 150-151: “12 trained with manual conditioning and 21 with the automated conditioning procedure”. Why this numerical difference?

-Line 214-216: Clarify this sentence

-Table 3: “Subjects.” , Please delete “.”

-Line 674:  Delete “increased”.

Sincerely

Author Response

RESPONSE TO REFEREE 3

The paper by E. Gatto and colleagues is undoubtedly a very good and interesting paper. A scrupulous description of methods and a large collection of information which are reported and expressed in an excellent way. I have only few suggestions.

-Please format the manuscript based on the author's guide. Methods together and all results together can help the readers.

-Is there a reason you only used female specimens?

-Lines 150-151: “12 trained with manual conditioning and 21 with the automated conditioning procedure”. Why this numerical difference?

-Line 214-216: Clarify this sentence

-Table 3: “Subjects.” , Please delete “.”

-Line 674:  Delete “increased”.

##Thank you very much for your helpful comments. Following the author’s guide, we modify the format of each sections to increase the legibility of our manuscript.

## We addressed your comments and, in particular, we justified the difference in sample size among experiments (Lines 178-187) and explained why we used only females (Lines 168-170). Finally, we carefully considered your minor suggestions through the manuscript.
